The landscape configuration of zoonotic transmission of Ebola virus disease in West and Central Africa: interaction between population density and vegetation cover

Walsh Michael G. 1 thegowda@gmail.com michael.walsh@downstate.edu
Haseeb MA 1 2
1 Department of Epidemiology and Biostatistics, School of Public Health, State University of New York , Downstate Medical Center, Brooklyn, NY , USA
2 Departments of Cell Biology, Pathology and Medicine, College of Medicine, State University of New York, Downstate Medical Center , Brooklyn, NY , USA
Nock Nora
Electronic publication date: 2015 Jan 20
Publication date: 2015
Volume: 3
Electronic Location ID: e735
Received 2014 Oct 29; Accepted 2015 Jan 2
Copyright: © 2015 Walsh and Haseeb
Copyright year: 2015
Copyright holder: Walsh and Haseeb
License: This is an open access article distributed under the terms of the Creative Commons Attribution License, which permits unrestricted use, distribution, reproduction and adaptation in any medium and for any purpose provided that it is properly attributed. For attribution, the original author(s), title, publication source (PeerJ) and either DOI or URL of the article must be cited.
License URL: https://creativecommons.org/licenses/by/4.0/

Keywords: Ebolavirus, Ebola, Epidemiology, Landscape epidemiology, Infection ecology, Spatial epidemiology, Zoonotic disease, Spillover

Funding: The authors declare there was no funding for this work.

==============================
Ebola virus disease (EVD) is an emerging infectious disease of zoonotic origin that has been responsible for high mortality and significant social disruption in West and Central Africa. Zoonotic transmission of EVD requires contact between susceptible human hosts and the reservoir species for Ebolaviruses, which are believed to be fruit bats. Nevertheless, features of the landscape that may facilitate such points of contact have not yet been adequately identified. Nor have spatial dependencies between zoonotic EVD transmission and landscape structures been delineated. This investigation sought to describe the spatial relationship between zoonotic EVD transmission events, or spillovers, and population density and vegetation cover. An inhomogeneous Poisson process model was fitted to all precisely geolocated zoonotic transmissions of EVD in West and Central Africa. Population density was strongly associated with spillover; however, there was significant interaction between population density and green vegetation cover. In areas of very low population density, increasing vegetation cover was associated with a decrease in risk of zoonotic transmission, but as population density increased in a given area, increasing vegetation cover was associated with increased risk of zoonotic transmission. This study showed that the spatial dependencies of Ebolavirus spillover were associated with the distribution of population density and vegetation cover in the landscape, even after controlling for climate and altitude. While this is an observational study, and thus precludes direct causal inference, the findings do highlight areas that may be at risk for zoonotic EVD transmission based on the spatial configuration of important features of the landscape.

Introduction

Ebola virus disease (EVD) first emerged in 1976 in what are today the Democratic Republic of the Congo and South Sudan. Over the course of the last 4 decades there have been a total of 24 outbreaks in 10 countries in Central and West Africa. The case fatality associated with EVD outbreaks is typically high (mean = 65%; range = 32%–90%) causing fear and social disruption when they occur. Never has this been more apparent than during the 2014 EVD outbreak in West Africa, which is currently underway. This outbreak is the largest documented EVD event in history and has affected the populations of Guinea, Sierra Leone, Liberia, and Nigeria, with 10,141 confirmed, probable, and suspected cases and 4,922 deaths as of 24 October 2014 according to the Centers for Disease Control and Prevention case counts (CDC, 2014). Because of the increasing magnitude of the burden of EVD, locating sources of human outbreaks is an important public health priority. Ebolaviruses require two transmission processes for outbreaks to emerge. First, there must be a spillover event, which is defined as a zoonotic transmission from either the primary sylvan reservoir host (e.g., fruit bats) or from a secondary sylvan host for whom the virus is also pathogenic (e.g., non-human primates) (Allaranga et al., 2010). Second, once the spillover generates the index case(s), subsequent person-to-person spread follows the care of the affected individuals via contact with contaminated body fluids (Allaranga et al., 2010). As the upstream source of all subsequent human-to-human transmission, spillover is thus an important antecedent to widespread human outbreaks. Nevertheless, despite four decades of EVD outbreak management, as well as research into the epidemiology and infection ecology of Ebolaviruses, we still have yet to clearly delineate the biologic, physical, and social features of the landscape that defines zoonotic infection risk. More specifically, the collective experience of these twenty-four outbreaks have not yet fully revealed the landscape epidemiology of spillover events. Locating potential points of spillover by recognizable features in the landscape may prove useful in blocking zoonotic transmission to humans and thus preventing human outbreaks (which are notoriously difficult to control in the ensuing social disruption) before they happen. For example, ecologic, economic, and social disruption, which may lead to displaced human and animal populations with emerging pressures of deforestation and encroachment, have often been posited as determinants of EVD outbreaks (Bausch & Schwarz, 2014; Pourrut et al., 2005). Poverty is also an important driver of human–animal interaction (Brashares et al., 2011). However, while certainly ecologically plausible and supported by a few small local field investigations (Daszak, Cunningham & Hyatt, 2000; Georges et al., 1999; Leroy et al., 2004; Wolfe, 2005), this remains theoretical across much of the region where these outbreaks occur (Daszak, Cunningham & Hyatt, 2000; McCallum & Dobson, 1995). Moreover, while one recent investigation has provided valuable insight into the ecologic niche of EVD (Pigott et al., 2014), no studies to date have considered the collective experience of EVD as an explicitly spatial process. The current investigation sought to explore the role of human population density and vegetation cover on the spatial distribution and risk of zoonotic transmission of EVD. The historical and current experience of these EVD outbreaks from 1976 to 2014 was used to test the hypothesis that increasing human population density in forested landscapes would be associated with increased zoonotic transmission. This investigation does not assess human to human transmission dynamics and subsequent epidemics, but rather focuses specifically on isolated spillover events.

Methods

The PubMed database maintained by the National Library of Medicine was queried for all primary reports of outbreaks of Ebola virus disease (EVD) beginning with the initial outbreaks in the Democratic Republic of the Congo (previously Zaire) and South Sudan (previously Sudan) in 1976 and ending with the 2014 West Africa outbreak, which began in Guinea in 2013, and the 2014 outbreak in the Democratic Republic of the Congo. In addition, all archives of the World Health Organization’s (WHO) Weekly Epidemiological Record and other WHO reports were queried to identify any additional EVD outbreaks or individual cases that were not represented in the peer-reviewed scientific literature. A total of 36 reports were obtained (peer-reviewed manuscripts = 27; WHO reports = 9). From these reports, the index case(s) was identified from each outbreak and the geographic location recorded at the village level. Each of the 24 outbreaks, with their 32 distinct spillover events and locations, and all references are listed in Table S1. The locations of zoonotic transmission events were assigned geographic coordinates using longitude and latitude as the coordinate reference system and obtained in Google Maps and cross referenced with Open Street Map [ openstreetmap.org]. Only 21 of the 24 zoonotic EVD outbreaks (or 29 of 32 unique spillover events) were included in this analysis. Three spillover events were not considered in this analysis because either the precise location was not identified in the outbreak reports (Luwero District, Uganda, 2012–2013 and Isiro, DRC 2012), or could not be geolocated (Nakisamata, Uganda 2011). Given that this investigation is attempting to understand the occurrence of EVD spillover as a point process, it was deemed inappropriate to treat these three events as polygons in the analysis.

Table 1 Adjusted relative risks and 95% confidence intervals and p-values for the associations between zoonotic transmission of Ebola virus disease and features of the landscape.

These measures of association are derived from an inhomogeneous Poisson model of the point process of zoonotic transmission events with interaction between population density and Maximum Green Vegetation Fraction (MGVF).

Landscape factor	Relative risk*	95% confidence interval	P-value	
Population density (100 persons/km2)	0.98	0.97–0.99	<0.0001	
Mean Green Vegetation Fraction (MGVF)(%)	0.99	0.94–1.05	>0.1	
Mean temperature (C°)	0.93	0.88–0.98	<0.001	
Mean precipitation (cm)	1.00	0.99–1.001	>0.1	
Altitude (10 m)	0.96	0.94–0.99	<0.01	
Population density: MGVF interaction	1.0002	1.0001–1.0003	<0.0001	
Notes.

* The relative risks reflect the percent change in zoonotic transmission risk associated with each unit change (as listed in parentheses) in the corresponding landscape factor.

Climate data were obtained from the bioclimatic data maintained by the WorldClim Global Climate database (WorldClim—Global Climate Data, 2014). Specifically, the annual mean temperature and annual mean precipitation from 1950 to 2000 were each extracted as 30 arc second resolution rasters (Hijmans et al., 2005). A raster for altitude at the same resolution was also obtained. Each pixel in these rasters represents the annual mean temperature, annual mean precipitation, and altitude for that approximately 1 square kilometer point on the Earth’s surface.

Similarly, we acquired a raster for the MODIS-based Maximum Green Vegetation Fraction (MGVF) from the United States Geologic Survey’s Land Cover Institute (USGS Land Cover Institute, 2012). This raster describes the percentage of green vegetation cover per pixel and is a function of the normalized difference vegetation index (Broxton et al., 2014). The MGVF raster resolution is, again, approximately 1 kilometer squared.

Finally, human population density was obtained from the Socioeconomic Data and Applications Center (SEDAC), which is part of the National Aeronautics and Space Agency’s Earth Observing System Data and Information System (Socioeconomic Data and Applications Center, 2014). Population density is represented by a 30 arc-second raster derived from the Global Rural-Urban Mapping Project estimates for the 2000 population, where each pixel represents the population density for that approximately 1 square kilometer point on the Earth’s surface (Balk et al., 2006).

Statistical methods

The 29 geolocated zoonotic EVD spillover events were modeled as a point process (Baddeley & Turner, 2000) across the tropical belt of narrow latitude and wide longitude in West and Central Africa. The specific geographic extent of the investigation was latitude 40°S, 35°N, and longitude 20°W, 60°E.

First, as a null model indicating complete spatial randomness, the outbreaks were considered as a homogeneous Poisson process with conditional intensity, λu,X=β

where u represents the set of locations corresponding to the pattern of points, X, and β is the intensity parameter. With this formulation, the expected number of points (i.e., intensity) in any subregion of the larger geographic extent is simply proportional to the area of that subregion (Baddeley & Turner, 2000).

Subsequently, the model representing a homogeneous Poisson process was compared to a model representing an inhomogeneous Poisson process, which indicates a spatial dependency in the pattern of zoonotic transmisison, and has conditional intensity, λu,X=βu

and now shows the intensity as a function of the location, u, of the points. Given the better model fit of the inhomogeneous Poisson process (see ‘Results’), the investigation proceeded under the assumption that the intensity was spatially dependent and, thus, attempted to identify those landscape features that were associated with the spatial distribution of zoonotic transmission events in West and Central Africa. These features were included in the inhomogeneous point process model as spatially explicit covariates with conditional intensity, λu,X=ρZu

where ρ is the function describing the association between the point intensity and the set of covariates Z at location u.

Six covariates corresponding to the landscape features of interest were considered in this analysis: population density, MGVF, mean temperature, mean precipitation, altitude, and an interaction covariate between population density and MGVF. Population density is presented in the results in increments of 100 persons/km2, and altitude is presented in 10 m increments. The association between the intensity of zoonotic EVD transmission events and each covariate was assessed via the relative risks derived from the regression coefficients of the inhomogeneous Poisson model. First, a homogeneous Poisson “null” model was fit and compared to an inhomogeneous model, which specified only spatial trend. Subsequently, an inhomogeneous model with the covariates described above was fitted to the EVD spillover events to identify possible sources of spatial dependency. The covariates in the full model assess the independent association between intensity and each landscape factor adjusted for all other factors, while simultaneously assessing the interaction between population density and MGVF. We did also consider both temperature range and isothermality in our model. However, these contributed to multicollinearity with mean temperature leading to model instability and so, as these were not significantly associated with EVD spillover events (RR = 1.0, 95% CI [0.99–1.00] for both together and independently), these were not included in the final analysis. The R language was used for all analyses (http://www.r-project.org/). The ppm function in the spatstat package was used for the point process model, and the valid.ppm function was used to verify that the fitted models specified well-defined point processes (Baddeley & Turner, 2005). The R code for these models is presented in File S1.

Results

The geographic distribution of the 29 precisely geolocated zoonotic transmission events that occurred in West and Central Africa between 1976 and 2014 are depicted in the map in Fig. 1. Comparison of the homogeneous and inhomogeneous Poisson process models based on the likelihood ratio test (p < 0.05) suggests that zoonotic EVD transmission is spatially dependent.

Figure 1 The distribution of zoonotic Ebola virus disease transmission events (red dots) across West and Central Africa.

Table 1 presents the regression coefficients from the inhomogeneous Poisson process model, which represents the adjusted measures of association between zoonotic EVD and each covariate. That is, the measure of association for each landscape factor is adjusted for all others in the model. Both population density (RR = 0.98, 95% CI [0.97–0.99]) and MGVF (RR = 0.99, 95% CI [0.94–1.05]) were inversely associated with the spatial distribution of zoonotic transmission events, wherein increasing population density or vegetation cover, respectively, corresponded to decreasing spillover risk. However, given that this model assessed the association between zoonotic EVD and the landscape factors population density and MGVF, each as modifying the other, we must also consider the significant interaction between them (RR = 1.0002, 95% CI [1.0001–1.0003]) to arrive at the correct interpretation. The interaction term describes how the relationship between zoonotic EVD and each of the two landscape factors changes at different levels of the other. For example, at a population density of 0 persons/km2 the association between vegetation cover is simply the RR for MGVF, where each percentage increase in cover corresponded to a 2% decrease in spillover risk. However, each 100 persons/km2 increase in population density alters the association between MGVF and zoonotic EVD by a factor of 2% (RR = 1.02 × 0.99 = 1.0098), which corresponds to a diminishing protective effect of vegetation cover as the population density for that area increases. Indeed, vegetation cover is no longer protective at a population density of just 100 persons/km2. A threshold of population density is reached at 200 persons/km2 after which zoonotic transmission risk is greater than 1% with each 1% increase in vegetation cover (RR = 1.022 × 0.99 = 1.03). Similar effect modification of population density by MGVF is implied. In addition, temperature and altitude were both associated with zoonotic EVD with each 1° increase in temperature and 10 m increase in altitude corresponding to a 7% and 4% decrease in risk, respectively. The point process was verified as being well-defined for the fitted model, and the likelihood ratio test comparing the interaction model to a no interaction model suggested that the former was the better fit (p < 0.01).

Figure 2 displays maps of the distribution of temperature, altitude, vegetation cover, and human population density across the African continent with the zoonotic transmission event points superimposed. All landscape factors show some level of heterogeneity across much of the range of zoonotic EVD transmission, but population density appears much more heterogeneous in space. Figure 3 maps the predicted intensity of spillover as a spatially dependent function of the landscape features in the inhomogeneous Poisson model. Red areas in the map represent areas where spillover events are predicted by the model. The map predicts spillover in areas of Gabon, the Republic of Congo, Uganda, and some smaller regions of the DRC, all countries which have seen significant outbreaks in the past. However, there are also some areas with predicted intensity in countries without previous spillover events such as Rwanda, Burundi, Cameroon, the Central African Republic, Nigeria, Ghana, and Cote d’Ivoire. It is important to note that the predicted spillover in all of these countries is highly area specific, with risk being determined by the spatial configuration of population density, vegetation cover, temperature, and altitude.

Figure 2 The distribution of mean annual temperature, altitude, Maximum Green Vegetation Fraction, and population density across the African continent with the distribution of zoonotic Ebola virus disease transmission events overlaid (red/blue dots).

In the map of population density, darker shades indicate higher population density, while areas of white indicate a population density less than 10 persons per km2. These maps depict the raster data distributions of the predictor variables and the observed point process of EVD spillover, rather than modeled associations.

Figure 3 Predicted intensity of zoonotic Ebola virus disease transmission in West and Central Africa based on the modeled inhomogeneous Poisson point process.

Areas of predicted zoonotic transmission events are depicted in red, while the observed points of zoonotic transmission are depicted by blue dots.

Discussion

This study explored the spatial dependence of 29 known EVD spillover events on specific landscape features within the geographic range where these events have occurred on the African continent. The interaction between human population density and vegetation cover was the single most important feature associated with the occurrence of zoonotic transmission. In the absence of population pressure, increasing vegetation cover was associated with decreased risk of zoonotic transmission. However, as population density increased across geographic space in the presence of higher vegetation coverage, the protective effect of MGVF was reversed and ultimately a threshold is reached at 200 persons/km2, after which zoonotic transmission risk is greater than 1% with each 1% increase in vegetation cover. These results suggest that the interaction between human population density and sylvan habitat may create pressure in the landscape that open conduits to EVD spillover into human communities.

It has been postulated that human encroachment on sylvan habitats is an important mediating factor in the emergence of novel infectious diseases in humans (Daszak, Cunningham & Hyatt, 2000; Estrada-Peña et al., 2014; Wolfe, Dunavan & Diamond, 2007; Woolhouse & Gowtage-sequeria, 2005), and in the emergence of EVD in particular (Bausch & Schwarz, 2014; Wolfe, 2005). More specifically, the repurposing of sylvan landscapes for agricultural, industrial, and residential land use frequently coincides with altered population densities that result from concentrated abrupt events such as conflict, or from the more diffuse and insidious processes of inequitable expansion of global and local economies (Bausch & Schwarz, 2014; Wolfe, Dunavan & Diamond, 2007). Typically, varying levels of forest fragmentation then accrue. This fragmentation often presents anthropogenic ecotones that have the potential to act as conduits for human–animal interaction where, previously, such contact was blocked by intact sylvan ecosystems (Smith & Smith, 2001). Moreover, shifting forms of land use among isolated populations, such as the location and extent of subsistence hunting, are often driven by poverty and inequitable economic development (Brashares et al., 2011). As such, where nascent human–animal interactions facilitate the transmission of novel pathogens from sylvan reservoirs to incidental human hosts, outbreaks of emerging infections may follow (Georges et al., 1999; Leroy et al., 2004; Wolfe, 2005). Nevertheless, very few investigations have attempted to measure the specific association between human influence in the landscape and the spillover of Ebolaviruses from animal to human populations. The current investigation was the first to attempt to quantify zoonotic transmission risk as spatially dependent on human–forest interaction across the complete zoonotic EVD experience in West and Central Africa. In one excellent recent study, investigators used an ecologic niche approach to identify the geographic range of Ebolavirus presence, and thus recognize areas of potential spillover to humans (Pigott et al., 2014). Their species distribution models used climate data, land cover, and the observed presence of presumed Ebolavirus reservoir bat species to model the presence or absence of Ebolaviruses based on all documented human and animal EVD outbreaks. The result is a nice delineation of the ecologic niche of Ebolaviruses across sub-Saharan Africa, the extent of which largely coincides with the predicted risk in the current study. However, the ecologic niche of the former study did not directly describe the relationship between human pressure in specific landscapes and zoonotic transmission, nor did it model the zoonotic transmission events explicitly as a point process with spatially dependent intensity. As such the current study adds to the previous work by highlighting areas of particular zoonotic transmission risk due to the spatial configuration of the interaction between human population density and vegetation cover in the landscape. It is also worth noting that, by itself, vegetation cover was not strongly associated with EVD spillover, while population density was. Nevertheless, the relevance of vegetation cover is more apparent in its interaction with population density. We feel this also makes sense intuitively since, by itself, an area of intact forest with dense vegetation cover could conceivably either elevate or decrease risk. Presumably, one would be more likely to encounter the natural reservoir for an Ebolavirus while in a pristine sylvan habitat and, as such, it may appear that dense vegetation cover confers risk. However, such a habitat may also be less likely to have people moving through it, which may give the appearance of diminishing risk. Conversely an urban community with much less vegetation cover would presumably be less likely to harbor the reservoir species and more likely to suppress human–animal encounters, and thus higher population density may appear to absorb risk. However, the actual risk of spillover ultimately depends on the potential for those human–animal encounters, which would require a minimum presence of sylvan space and human presence in close proximity to that space. The results of the current study support such a relationship.

This study also identified an inverse relationship between zoonotic EVD transmission and both temperature and altitude. Increases in both temperature and altitude corresponded to lower spillover risk, which conforms to the previously mentioned study by Pigott et al. (2014) for altitude but not temperature. Increasing altitude may reduce the necessary human–animal encounters because of climatic changes (i.e., reduced temperature) or plant and animal species distribution changes that attend changes in elevation, or because of lower population density at altitude. It is unclear why increasing temperatures were associated with lower spillover intensity in this study. It may be that areas of higher mean temperature were also areas of lower minimum temperatures, as is the case in more northern latitudes entering into the African Sahel which, to date, has also demarcated the northern boundary of Ebolavirus transmission. Moreover, such a boundary may in fact be due to differences in the temperature range, or may simply be due to a lack of the reservoir species range extending that far north (Pigott et al., 2014).

There are some important limitations inherent in this study. First, the rasters for temperature and precipitation consisted of single composite measures over the period 1950 to 2000. In addition, the earliest available raster for MGVF was from 2001. As such, while these three rasters exhibited high spatial resolution (∼1 square kilometer), the temporal resolution was coarse, given that the former two were averaged over a 50 year time span and the latter was a single measurement. Nevertheless, we feel that the measures of temperature and precipitation in this study are more indicative of the general climate of the regions represented, and thus provide a more robust approach to controlling for background climate while assessing the impact of human influence. These climate measures would be expected to be less accurate for assessing more nuanced patterns in weather on the occurrence of zoonotic EVD transmission. In addition, the measure of MGVF, which was captured in this raster in 2001, was taken roughly at the median year (2002) of all EVD outbreaks and, as such, can be considered an approximation of green vegetation cover at the temporal midpoint of the zoonotic transmission point process. Again, while such a measure may not be sufficient for assessing the effect of complex change in vegetation cover over time, we feel that, as a measure of the central tendency of MGVF, it is an appropriate representation of the background vegetation cover while assessing human population density at the same time period. Second, the number of occurrences comprising the full history of documented EVD outbreaks in Africa is relatively small. As such, the sample size for this investigation is also small, with only 29 precisely geolocated spillover events available. Therefore, this collection of zoonotic events may not be representative of the total potential events in the region, and the model may overestimate risk for landscape features and geographic areas with a high relative occurrence and underestimate risk for landscape features and geographic areas with a low relative occurrence. Third, as described above in the Statistical Methods section, temperature range and isothermality were excluded from the model due to multicollinearity. As such, these are two climatic factors that were not included in the inhomogeneous Poisson model, and thus predictions of spillover risk are less reflective of climate nuance than they would be if these two factors were included. Nevertheless, as mentioned above, neither temperature range nor isothermality were significantly associated with EVD spillover events (RR = 1.0, 95% CI [0.99–1.00] for both) so we expect that their influence on predicted spillover risk is not dramatic. Finally, this is fundamentally an observational study, which precludes any direct causal interpretation of the findings. The associations described are just that: observational associations. They may suggest relationships between features of the landscape and zoonotic EVD risk, but they do not definitively identify what could be called true causal relationships. The latter can only be discerned by direct measurement of animals and humans in the specific locations where zoonotic transmission events are taking place. This will require extensive field investigation incorporating human and animal serology, habitat description at a much finer resolution and richer detail (i.e., describing not simply vegetation cover, but providing more precise descriptions of habitat parcels in terms of biologic and physical properties), and specific cultural and economic practices that bring humans into close proximity to reservoir species. Moreover, identification of external sources of local population movement and displacement may be particularly useful in understanding the ways in which population density interacts with sylvan landscapes to confer risk (Bausch & Schwarz, 2014). Such research is an expensive prospect indeed, but it is necessary if we are to generate sufficiently rich data to inform actionable prevention structures that have real potential to limit dangerous human–animal encounters and block zoonotic EVD transmission.

In conclusion, the current investigation identified a strong association between EVD spillover and population density and effect modification with vegetation cover. The results also suggest relationships with temperature and altitude, though these may be incidental to the as yet unknown range of the reservoir species. While these findings cannot be interpreted as causal due to the observational nature of the data, they do suggest that the specific landscape configuration of interaction between human populations and forested land may facilitate zoonotic EVD transmission.

Supplemental Information

Table S1 Description of all zoonotic Ebola virus disease outbreaks

Click here for additional data file.

Supplemental Information 2 EVD Data

Click here for additional data file.

File S1 Point process model R code

Click here for additional data file.

Additional Information and Declarations

Competing Interests

Author Contributions

The authors declare there are no competing interests.

Michael G. Walsh analyzed the data, wrote the paper, prepared figures and/or tables, reviewed drafts of the paper.

MA Haseeb wrote the paper, reviewed drafts of the paper.

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
