# Peer review of "The landscape configuration of zoonotic transmission of Ebola virus disease in West and Central Africa: interaction between population density and vegetation cover"

_PeerJ, doi:10.7717/peerj.735_

## Round 0.1 · original submission · Major Revisions

· Academic Editor

Major Revisions

Please address all of the issues raised by the reviewers paying particular attention to the need for better quantitative analyses to support the statements made in the results, discussion and conclusions. P-values should be provided to support the effect sizes and confidence intervals/standard errors.

Reviewer 1 ·

Basic reporting

Authors needs to explain the meaning of the "transmission risk" areas shown in Figure 3. How should the reader interpret red these areas?

Please include a P value next to each of the covariates shown in Table 1. The relative risks shown are too close to 1.0. I am concerned these estimates are not statistically significant? As the authors will agree, this is a very important issue that needs careful assessment.

Experimental design

none

Validity of the findings

I would like the authors to check the statistical significance of their findings.

Additional comments

Authors imprecisely refer to R0 as a level of "infectiousness" -- R0 is a measure of transmission potential. Moreover, R0 just refers to community averages and does not necessarily captures the variability at the individual level. It is important to note that superspreading events have played a significant role in the development of Ebola epidemics. Even in Nigeria, the index case infected 12 secondary cases (R0=12), but interventions measures quickly reduced the effective reproduction number below 1.0 (Fasina et al. Eurosurveillance 2014). It is also important to note that R0 is not a fixed quantity but varies geographically as a function of a number of factors including health care infrastructure. R0 is different in each of the three most affected countries in West Africa, and several studies have attempted to estimate R0 at the country level.

Reviewer 2 ·

Basic reporting

See comments below

Experimental design

See comments below

Validity of the findings

See comments below

Additional comments

The following comments are to be clarified
1. The present study on “The landscape configuration of zoonotic transmission of Ebola virus disease in West and Central Africa: Interaction between population density and vegetation cover”, is a good attempt, and observational study, but not research article. And the following comments of statements to be clarified.

2. The study result highlighted that climate variables and altitude has negative relationship, however, the explanation is crude and qualitative statement. And, it is to be recalculated because, the survival of virus and temperature and precipitation range, climate and altitude, and landscape environmental suitability are not studied systematically (temperature range /temperature interval is not defined).

3. The highlights of spatial relationship between the landscape environment given in the study that the linear relationship between (i) the human settlement clusters / population density and Ebola virus disease epidemics, (ii) vegetation cover and epidemics. It must be clarified that is it possible or likely to spread the Ebola virus disease epidemics in an area where the same environmental condition across the Africa continent / globe? The authors have to be considered these points, and to be studied the spatial relationship between the landscape environmental variables and the possibility of Ebola virus disease epidemics in the non-epidemic areas in other part of the affected country/ neighbouring countries, then only the results could be ascertained and considered.

4. The study of landscape environmental variables (vegetation indices) and the Ebola virus disease epidemics is so complex, therefore, a study must be made to reveal the risk factors of epidemics thoroughly, not a observational and qualitative statements.

Therefore, my reviewer conclusion and suggestion is that the paper should be revised and may be resubmitted for publication.

---

## Round 0.2 · Minor Revisions

· Academic Editor

Minor Revisions

The authors have addressed the majority of the comments previously raised. However, there area a few minor revisions that should be addressed as follows:

In Table 1 and in the Results Section: Please provide the relative risk (RR) and 95% CI of RR for more meaningful interpretations (in addition to the 1 unit change in the RR). For example, in the rebuttal letter, the authors mention an example for 10 m or 100 m increases in altitude. The authors should decide which unit change(s) are the most meaningful and provide the corresponding RR and 95% CIs. To the extent necessary, please also update any relevant statements in the Discussion section and conclusions.

The problems with multicolinearity should also be mentioned in the study limitations paragraph in the Discussion section.

Reviewer 1 ·

Basic reporting

Authors have adequately addressed my comments. I have not further comments.

Experimental design

no comments

Validity of the findings

no comments

---

## Round 0.3 · Minor Revisions

· Academic Editor

Minor Revisions

In the previous round of reviews, we commented that "In Table 1 and in the Results Section: Please provide the relative risk (RR) and 95% CI of RR for more meaningful interpretations (in addition to the 1 unit change in the RR). For example, in the rebuttal letter, the authors mention an example for 10 m or 100 m increases in altitude. The authors should decide which unit change(s) are the most meaningful and provide the corresponding RR and 95% CIs. To the extent necessary, please also update any relevant statements in the Discussion section and conclusions." The authors did address this for altitude (10 m) but they did not address this for the other parameters listed. To clarify further, we would like to see the results for the 1 unit change AND a meaningful XXX unit change for EACH "landscape feature" in Table 1 and the corresponding presentation and discussion of these results in the Results and Discussion sections, respectively.

---

## Round 0.4 · accepted · Accept

· Academic Editor

Accept

The authors points are well taken. Although I do not think having both RR estimates (for a 1 unit change and a meaningful or more interpretable unit change) would be confusing to the reader if properly documented, I will approve as is to prevent any further delays in the process.